# The cerebellum is involved in processing of predictions and prediction errors in a fear conditioning paradigm

Thomas Michael Ernst[1,2]*, Anna Evelina Brol[1], Marcel Gratz[2,3], Christoph Ritter[1], Ulrike Bingel[1], Marc Schlamann[4,5], Stefan Maderwald[2], Harald H Quick[2,3], Christian Josef Merz[6†], Dagmar Timmann[1,2†]*

[1]Department of Neurology, Essen University Hospital, Essen, Germany; [2]Erwin L. Hahn Institute for Magnetic Resonance Imaging, University of Duisburg-Essen, Essen, Germany; [3]High-Field and Hybrid MR Imaging, Essen University Hospital, Essen, Germany; [4]Institute of Diagnostic and Interventional Radiology and Neuroradiology, Essen University Hospital, Essen, Germany; [5]Department of Neuroradiology, University Hospital Cologne, Cologne, Germany; [6]Department of Cognitive Psychology, Institute of Cognitive Neuroscience, Ruhr University Bochum, Bochum, Germany

*For correspondence:
thomas.ernst@uk-essen.de (TME);
Dagmar.Timmann-Braun@uni-duisburg-essen.de (DT)

†These authors contributed equally to this work

Competing interests: The authors declare that no competing interests exist.

Reviewing editor: Sam McDougle,

**Abstract** Prediction errors are thought to drive associative fear learning. Surprisingly little is known about the possible contribution of the cerebellum. To address this question, healthy participants underwent a differential fear conditioning paradigm during 7T magnetic resonance imaging. An event-related design allowed us to separate cerebellar fMRI signals related to the visual conditioned stimulus (CS) from signals related to the subsequent unconditioned stimulus (US; an aversive electric shock). We found significant activation of cerebellar lobules Crus I and VI bilaterally related to the CS+ compared to the CS-. Most importantly, significant activation of lobules Crus I and VI was also present during the unexpected omission of the US in unreinforced CS+ acquisition trials. This activation disappeared during extinction when US omission became expected. These findings provide evidence that the cerebellum has to be added to the neural network processing predictions and prediction errors in the emotional domain.
DOI: https://doi.org/10.7554/eLife.46831.001

## Introduction

Cerebellar disease has long been known to result in disordered motor performance and motor learning (*Holmes, 1908*; *McCormick and Thompson, 1984*). Evidence has accumulated that cerebellar patients also present with various degrees of cognitive, emotional and behavioral abnormalities (*Schmahmann and Sherman, 1998*). Because the microscopic structure of the cerebellum is highly homogeneous, it is often assumed that the cerebellum performs one single neural operation (see Miall in *Caligiore et al., 2017*; *Popa et al., 2014*; *Sokolov et al., 2017*). The most popular current hypothesis states that the cerebellum acts as, or is part of, a predictive device (*Popa and Ebner, 2018* for recent review). In the motor domain, it is assumed that the cerebellum is crucially involved in prediction of the sensory consequences of motor commands thought to be achieved via internal models (*Bastian, 2006*; *Miall et al., 1993*; *Wolpert et al., 1998*). These internal models have to be constantly adapted because of a constantly changing inner and outer environment. Assumedly, the difference between the predicted and actual sensory outcome results in a sensory prediction error used to adapt the internal model and subsequent motor behavior. Although most studies have been performed in the motor domain, there is initial evidence that the cerebellum is involved in predictive

control in the cognitive domain (*Lesage et al., 2012*; *Lesage et al., 2017*; *Moberget et al., 2014*). The aim of the present study was to show that this assumption also applies to the emotional domain.

Fear conditioning was used as a model system because the cerebellum is involved in acquisition of learned fear responses (*Lange et al., 2015*; *Maschke et al., 2002*; *Ploghaus et al., 1999*; *Sacchetti et al., 2002*), and has known connections with several parts of the neural network underlying fear conditioning, including the limbic system (*Badura et al., 2018*; *Blatt et al., 2013*). Furthermore, prediction errors are thought to be the main drivers of associative fear learning (*Holland and Schiffino, 2016*; *Rescorla and Wagner, 1972*). In the fear conditioning literature, however, the possible role of the cerebellum in aversive prediction error processing has largely been ignored (*Apps and Strata, 2015*; *Tovote et al., 2015*). Previous studies focused on the role of the amygdala, insula, midbrain periaqueductal gray and striatum (*Boll et al., 2013*; *Li et al., 2011*; *Li and McNally, 2014*). We wanted to provide initial evidence that the cerebellum has to be added to the neural network processing predictions errors in learned fear responses.

During fear conditioning, participants learn to predict that the initially neutral conditioned stimulus (CS) is followed by an unpleasant unconditioned stimulus (US). As a result, fear responses are elicited already at the time of CS presentation. The initial occurrence of the US is unexpected and has been considered as an error signal (*Taylor and Ivry, 2014*). An event-related functional magnetic resonance imaging (fMRI) design allowed us to separate blood oxygenation level-dependent fMRI signals related to the CS, from signals related to the subsequent US. Participants learn within a very limited number of trials that the CS predicts the occurrence of the US, particularly if appropriate instructions are provided (*Atlas et al., 2016*; *Tabbert et al., 2011*). In case the cerebellum is involved in prediction of the US, cerebellar fMRI signals should be high during CS presentation. As soon as learning has occurred, the occurrence of the US is expected. Thus, if the hypothesis is correct that the cerebellum contributes to aversive prediction errors, cerebellar activation should be increased at unexpected omission of the US (because of a partial reinforcement schedule). During extinction, that is the repeated presentation of CS-only trials, omission of the US becomes expected and cerebellar fMRI signals at the time of US omission should decrease.

In accordance with the fMRI literature (*Lange et al., 2015*), we found cerebellar activations related to prediction of the US. In addition, marked cerebellar activation was present during unexpected omission of the US, which disappeared during extinction. Our findings are consistent with the hypothesis that the cerebellum is involved in processing of aversive predictions and prediction errors, and has to be added to the neural network underlying emotional associative learning.

## Results

While acquiring 7T fMRI data, 27 participants underwent a differential fear conditioning and extinction paradigm using visual stimuli as CS and an unpleasant, but not painful electrical shock as aversive US. *Figure 1* displays the experimental paradigm as well as the event blocks chosen for fMRI analysis. Behavioral parameters included self-reported ratings prior to and after the acquisition and extinction phases, and skin conductance responses (SCR) during the experiment. Five supplementary tables are provided in *Supplementary file 1*.

### Behavioral data

#### Questionnaires

##### Valence and arousal ratings

After habituation and prior to acquisition, there was no difference in (hedonic) valence and (emotional) arousal ratings of the CS+ and CS- (*Figure 2a*). After acquisition, valence of the CS+ was rated less pleasant than that of the CS-. Additionally, arousal to the CS+ was rated higher than that to the CS-. Differences between CS+ and CS- ratings remained after extinction, a finding that has been reported as resistance to extinction in evaluative conditioning research (e.g. *Blechert et al., 2008*; *Vansteenwegen et al., 2006*). ANOVA with repeated measures showed a significant difference within stimulus types and phases (pre-acquisition, post-acquisition, post-extinction) considering both valence and arousal (main effects: all p<0.002). Valence and arousal ratings differed between stimulus type and phases (interaction stimulus type × phase: valence: $F_{2,42}$ = 14.95, p<0.001;

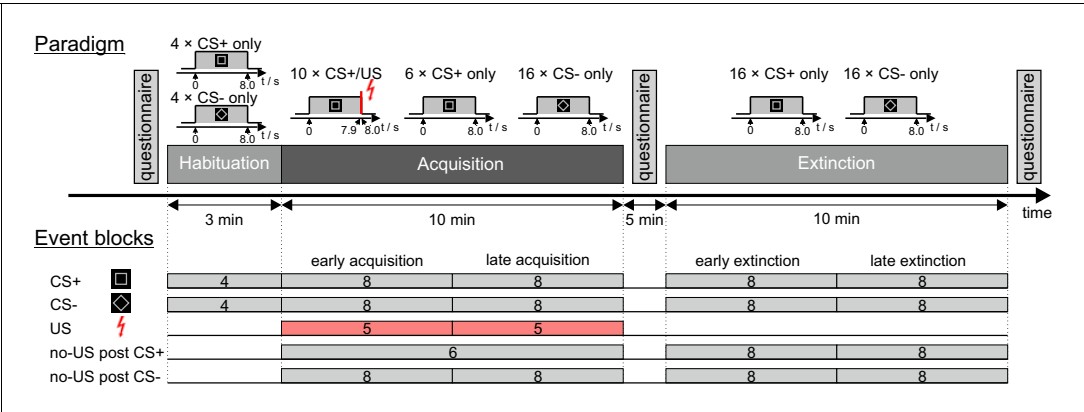

**Figure 1.** Experimental paradigm and overview of individual events. CS = conditioned stimulus; US = unconditioned stimulus. For further details see text.

DOI: https://doi.org/10.7554/eLife.46831.002

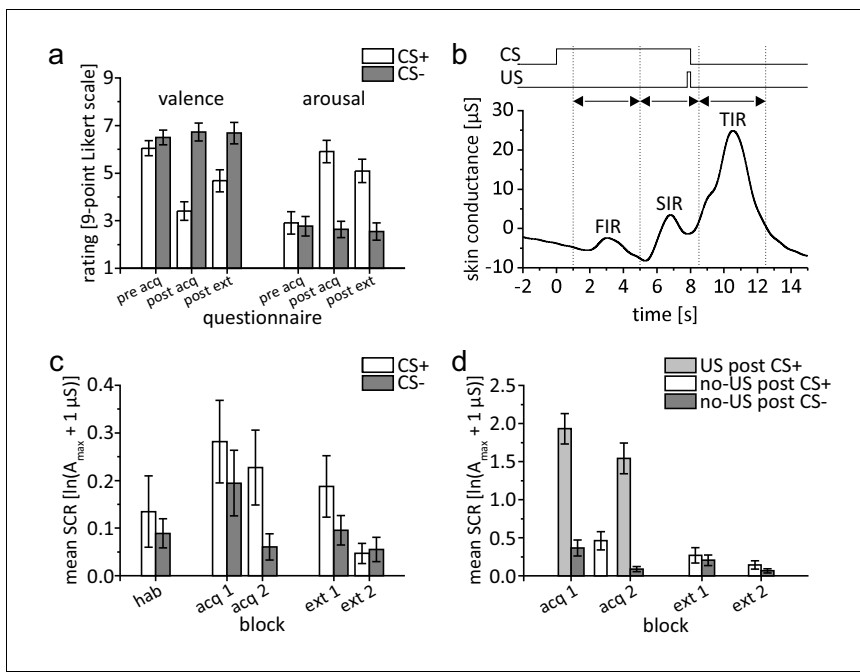

**Figure 2.** Behavioral data. (a) Group mean valence and arousal ratings for CS+ and CS- during acquisition and extinction. (b) Example of bandpass filtered individual SCR in a paired CS+/US trial depicting response interval windows and displaying a distinct response in each interval. (c) Group mean SIR. (d) Group mean TIR. Please note the different scales of the y-axis used for illustration purposes. Error bars represent standard errors of the mean. acq 1, acq 2 = early and late acquisition; CS = conditioned stimulus; ext 1, ext 2 = early and late extinction; FIR = first interval response; hab = habituation; SCR = skin conductance response; SIR = second interval response; TIR = third interval response; US = unconditioned stimulus.

DOI: https://doi.org/10.7554/eLife.46831.003

The following source data and figure supplement are available for figure 2:

**Source data 1.** Mean SCR data.
DOI: https://doi.org/10.7554/eLife.46831.005

**Figure supplement 1.** SCR related to CS presentation: FIR and SIR in comparison.
DOI: https://doi.org/10.7554/eLife.46831.004

arousal: $F_{2,42}$ = 15.30, p<0.001). Post hoc tests showed a significant difference between stimulus types after acquisition and after extinction (all p≤0.005; paired $t$-test), but not prior to acquisition (valence: p=0.781, arousal: p=0.125).

## US unpleasantness and CS-US contingency

After acquisition, the mean US unpleasantness rating was 6.9 (SD = 1.4) on a 9-point scale from 'not unpleasant' to 'very unpleasant'. All participants were aware of CS-US contingencies after the acquisition phase: The mean estimated probability that a CS+ was followed by an US was 70.0% (SD = 13.0%). All but one participant estimated a 0% probability of a CS- being followed by a US, with the remaining participant stating a 10% chance. Participants stated that they became aware of CS-US contingencies after 2.9 min (SD = 1.2 min), or 2.6 (SD = 0.8) US events.

## SCR

During habituation, second interval response (SIR) was not significantly different in CS+ and CS- trials ($t_{21}$ = 0.708, p=0.487; paired $t$-test) (*Figure 2c*). During fear acquisition, SIR was significantly higher in CS+ trials compared to CS- trials (*Figure 2c*). This difference was most pronounced in the second half of the acquisition phase. ANOVA with repeated measures showed a significant main effect of stimulus type (CS+ vs. CS-; $F_{1,21}$ = 5.182, p=0.033) and block (early vs. late; $F_{1,21}$ = 5.589, p=0.028). The stimulus type by block interaction was not significant ($F_{1,21}$ = 1.409, p=0.249).

During fear extinction, SIR related to the CS+ declined. In the second half of extinction the difference between CS+ and CS- trials vanished (*Figure 2c*). The main effects of stimulus type (CS+ vs. CS-; $F_{1,21}$ = 2.923, p=0.102) and block (early vs. late; $F_{1,21}$ = 3.930, p=0.061) were not significant. ANOVA with repeated measures showed a significant stimulus type by block interaction ($F_{1,21}$ = 5.035, p=0.036). Post hoc testing showed a significant difference between stimulus types during early ($t_{21}$ = 2.24, p=0.036), but not during late extinction ($t_{21}$ = −0.36, p=0.723).

Findings concerning first interval response (FIR) were comparable to SIR and are summarized in *Supplementary file 1* Table 1 and *Figure 2—figure supplement 1*.

SCRs in the unconditioned response (UR) window (i.e. the third interval response, TIR) were significantly higher in paired CS+ trials (US post CS+) compared to CS- trials (no-US post CS-) ($F_{1,21}$ = 93.70, p<0.001) indicating a successful increase in SCR towards the electric shock (*Figure 2d*). Block effect was significant (early vs. late; $F_{1,21}$ = 21.97, p<0.001) revealing higher UR during early compared to late acquisition. The block by stimulus type interaction was not significant ($F_{1,21}$ = 0.75, p=0.396). TIR was also significantly higher in unpaired CS+ trials (no-US post CS+) compared to CS- trials (no-US post CS-) during late acquisition ($t_{21}$ = 3.72, p=0.001) but not early acquisition ($t_{21}$ = 1.74, p=0.096), showing a higher US expectancy in US omission trials. TIR in unpaired CS+ trials was significantly smaller compared to TIR in paired CS+ trials (paired $t$-tests, all $p$ values < 0.001). During extinction, TIR was not significantly different comparing stimulus types (no-US post CS+ vs. no-US post CS-; $F_{1,21}$ = 3.46, p=0.077), blocks (early vs. late; $F_{1,21}$ = 3.72, p=0.067) or their interaction (stimulus type by block; $F_{1,21}$ = 0.02, p=0.878).

Taken together, we could show successful fear acquisition and extinction as well as a response towards the presentation and omission of the US during fear acquisition.

## fMRI data

We were interested in cerebellar activations related to i) the presentation, ii) the prediction, and iii) the omission of the aversive electrical stimulation (that is, the US). The focus of the data analysis was cerebellar activations. In addition, exploratory data on whole brain analysis are presented. Activation clusters are reported which are significant after application of threshold-free cluster-enhancement (TFCE) at p<0.05 familywise error (FWE) corrected level in all cases but conjunction analysis. Conjunction analysis results are reported at the level of p<0.05 FWE without TFCE.

### Cerebellar analysis
#### Cerebellar activation related to presentation of the aversive stimulus [contrast 'US post CS+ > no US post CS-']

Widespread cerebellar activation was observed within the cerebellar vermis and both cerebellar hemispheres (*Figure 3a*; see also *Table 1*). Most prominent differential activations were found in the

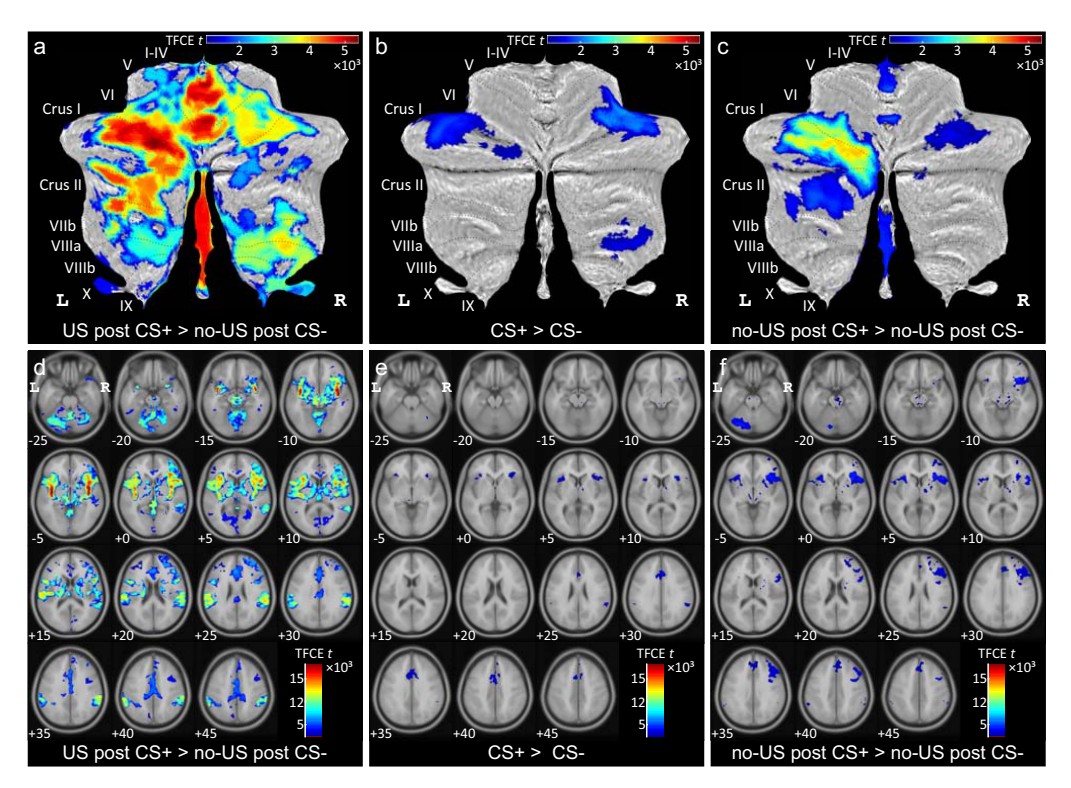

**Figure 3.** Diffential activations during fear acqusition. (**a–c**) Differential cerebellar activations during fear acquisition in SUIT space projected on a cerebellar flatmap (*Diedrichsen and Zotow, 2015*). (**d–f**) Corresponding differential whole brain activations in MNI normalized space. All contrasts collapsed over early and late acquisition blocks and calculated using TFCE and familywise error correction (p<0.05). CS = conditioned stimulus; L = left; MNI = Montreal Neurological Institute standard brain; R = right; SUIT = spatially unbiased atlas template of the cerebellum; TFCE = threshold-free cluster-enhancement; US = unconditioned stimulus.

DOI: https://doi.org/10.7554/eLife.46831.007

The following figure supplements are available for figure 3:

**Figure supplement 1.** Changes in differential cerebellar activation across acquisition and extinction blocks based on F-tests.
DOI: https://doi.org/10.7554/eLife.46831.008

**Figure supplement 2.** Effect of physiological denoising and smoothing kernel.
DOI: https://doi.org/10.7554/eLife.46831.009

anterior and posterior vermis (local maxima in lobules I-IV, V) and the left hemisphere (that is ipsilateral to the presentation of the US; local maximum in Crus I). Activation was not confined to the cerebellar cortex, but extended into the cerebellar nuclei (including dentate, interposed and fastigial nuclei bilaterally).

To assess changes in differential cerebellar activation across the two acquisition blocks (early and late) an *F*-test based on second level within-subject ANOVA was calculated. No significant main effect of block was observed during acquisition (at p<0.05 FWE corrected level, based on the TFCE statistic).

### Cerebellar activation related to the prediction of the aversive stimulus [contrast 'CS+ > CS-']

Cerebellar activation related to the CS+ was significantly higher in the lateral cerebellar hemispheres compared to activation related to the CS- (*Figure 3b*). Cerebellar activation was present in the more lateral parts of lobules VI and Crus I bilaterally (see also *Table 1*). Additional differential activation was present in lobules VIIIa and VIIIb in the right cerebellar hemisphere. During extinction, cerebellar activation related to the CS+ was not significantly different from activation related to the CS- (at p<0.05 FWE corrected level, based on the TFCE statistic).

**Table 1.** Cerebellar activations during acquisition and extinction.

Displayed are all clusters of ≥20 mm³. In each cluster, up to three maxima are listed separated by ≥8 mm. Corresponding activations for whole brain analysis are summarized in *Supplementary file 1* Table 5.

| Index | Location (lobule) | Side | SUIT coordinates/mm | | | Cluster size/ mm³ | $p_{FWE}$ | TFCE |
|---|---|---|---|---|---|---|---|---|
| *A) US post CS+ > no US post CS-: acquisition t-test, TFCE, p<0.05, FWE corr.* | | | | | | | | |
| 1 | Extended cluster | | left VI (8390), white matter (7950), left Crus I (7889), right VI (6404), right V (4250), left Crus II (4223), left V (4085), right Crus I (3457), right I-IV (2761), left I-IV (2529), right VIIIa (2432), right VIIIb (2244), left VIIb (1602), left VIIIb (1583), left VIIIa (1536), right VIIb (1467), vermal VI (1368), right IX (1330), vermal VIIIa (1307), right Crus II (1034), right dentate nuc. (921), vermal IX (804), left dentate nuc. (713), left IX (628), vermal VIIIb (474), vermal VIIb (236), right X (168), vermal Crus II (162), vermal X (120), left interposed nuc. (86), left X (70), right interposed nuc. (69), left fastigial nuc. (23), vermal Crus I (21), right fastigial nuc. (19) | | | | | |
| | Crus I | Left | −26 | −74 | −27 | 72355 | 0.001 | 5386.8 |
| | I-IV | Left | 0 | −53 | −24 | | 0.001 | 5373.2 |
| | V | Left | -3 | −62 | −23 | | 0.001 | 5032.2 |
| 2 | IX | Left | -5 | −47 | −51 | 39 | 0.025 | 1592.2 |
| 3 | IX | Right | 7 | −49 | −61 | 117 | 0.034 | 1435.8 |
| *b) CS+ > CS-: habituation t-test, TFCE, p<0.05 FWE corr.* | | | | | | | | |
| | no significant voxels | | | | | | | |
| *c) CS+ > CS-: acquisition t-test, TFCE, p<0.05, FWE corr.* | | | | | | | | |
| 1 | Extended cluster | | right Crus I (1506), right VI (1481), white matter (23), right V (16) | | | | | |
| | VI | Right | 35 | −50 | −31 | 3027 | 0.004 | 2256.6 |
| | VI | Right | 33 | −60 | −26 | | 0.004 | 2174.7 |
| | Crus I | Right | 40 | −57 | −32 | | 0.005 | 2082.8 |
| 2 | Extended cluster | | left Crus I (1658), left VI (727) | | | | | |
| | Crus I | Left | −44 | −56 | −33 | 2385 | 0.006 | 1911.7 |
| | Crus I | Left | −36 | −53 | −33 | | 0.007 | 1851.3 |
| | Crus I | Left | −41 | −64 | −31 | | 0.014 | 1629.8 |
| 3 | Extended cluster | | right VIIIa (287), right VIIIb (283), white matter (36), right VIIb (2) | | | | | |
| | VIIIb | Right | 28 | −48 | −49 | 608 | 0.019 | 1495.2 |
| | VIIIb | Right | 22 | −54 | −48 | | 0.020 | 1483.1 |
| | VIIIa | Right | 29 | −58 | −47 | | 0.037 | 1263.4 |
| 4 | Crus I | Left | −17 | −76 | −29 | 264 | 0.036 | 1278.4 |
| 5 | Crus I | Left | −34 | −75 | −25 | 46 | 0.047 | 1163.4 |
| *d) CS+ > CS-: extinction t-test, TFCE, p<0.05 FWE corr.* | | | | | | | | |
| | no significant voxels | | | | | | | |
| *e) no-US post CS+ > no US post CS-: acquisition t-test, TFCE, p<0.05, FWE corr.* | | | | | | | | |
| 1 | Extended cluster | | left Crus I (7688), left VI (4023), left Crus II (3373), white matter (1741), right I-IV (580), left VIIb (541), left dentate nuc. (474), left I-IV (472), vermal VIIIb (226), vermal IX (200), right interposed nuc. (163), vermal VIIIa (159), right dentate nuc. (92), left interposed nuc. (73), right V (70), left V (41), left IX (34), right fastigial nuc. (31), left VIIIa (30), vermal VI (9), right IX (9), vermal Crus I (8), left fastigial nuc. (8), vermal Crus II (2) | | | | | |
| | Crus I | Left | −17 | −78 | −25 | 20047 | <0.001 | 4010.8 |
| | VI | Left | −25 | −73 | −26 | | <0.001 | 3912.9 |
| | Crus I | Left | −41 | −68 | −29 | | 0.001 | 3633.1 |
| 2 | Extended cluster | | right Crus I (1313), right VI (750), white matter (66) | | | | | |
| | VI | Right | 30 | −68 | −27 | 2129 | 0.015 | 1484.6 |
| | VI | Right | 25 | −73 | −22 | | 0.018 | 1422.4 |
| | Crus I | Right | 45 | −65 | −27 | | 0.019 | 1384.5 |
| 3 | Crus II | Right | 15 | −79 | −33 | 42 | 0.047 | 1079.1 |
| *f) no-US post CS+ > no US post CS-: extinction t-test, TFCE, p<0.05, FWE corr.* | | | | | | | | |
| 1 | Crus I | Left | −14 | −72 | −35 | 273 | 0.016 | 1416.9 |

DOI: https://doi.org/10.7554/eLife.46831.006

*F*-tests revealed no significant block effects (early vs. late) neither during acquisition nor during extinction (at p<0.05 FWE corrected level, based on the TFCE statistic). The main effect of block across all four blocks (that is early and late acquisition, early and late extinction) revealed two clusters in the lateral cerebellum with local maxima in left lobule Crus I and right lobule VI (*Table 1*; *Figure 3—figure supplement 1a*). As can be seen from mean $\beta$ values of both clusters across blocks (see insert in *Figure 3—figure supplement 1a*), differential activation in the two clusters decreased during extinction compared to acquisition.

## Cerebellar activation related to the omission of the aversive stimulus [contrast 'no-US post CS+ > no US post CS-']

During acquisition, significant differential activation related to the (unexpected) omission of the US was found in the cerebellar hemispheres and the vermis (*Figure 3c*). Activation at the time of the expected US in unpaired CS+ trials compared to CS- trials was most prominent in the left cerebellar hemisphere with local maxima in lobules Crus I and VI (*Table 1*). Additional activation was present in the right hemisphere (local maxima in lobules Crus I and VI) and the vermis. Vermal activation was found in the anterior vermis (lobules I-IV, V) and the posterior vermis (lobules VIIb-IX). Activation extended into the cerebellar nuclei (including dentate, interposed and fastigial nuclei bilaterally). During extinction, cerebellar activation related to the (expected) omission of the US strongly decreased. Only one smaller cluster remained in more medial parts of left Crus I (*Table 1*; *Figure 3— figure supplement 1c*).

*F*-tests revealed no significant block effects (early vs. late) neither during acquisition nor during extinction (at p<0.05 FWE corrected level, based on the TFCE statistic). The main effect of block across all four blocks (that is early and late acquisition, early and late extinction) revealed a large cluster in the left hemisphere, primarily within lobule Crus I with some extension to lobule VI and Crus II (*Table 1*; *Figure 3—figure supplement 1b*). As can be seen from mean $\beta$ values across blocks (insert in *Figure 3—figure supplement 1b*), differential activation decreased during extinction compared to acquisition.

## Reanalyses of cerebellar activations

Reanalyses of the three main contrasts in the acquisition phase were performed without physiological denoising (see *Figure 3—figure supplement 2a-c*) and based on un-smoothed functional data (see *Figure 3—figure supplement 2g–h*). By omitting physiological denoising as well as by omitting smoothing cerebellar activation, patterns were essentially unchanged.

## Comparison of cerebellar areas related to presentation, prediction and omission of the aversive stimulus

Conjunction analyses were performed to reveal areas of cerebellar activation which were common to the presentation, the prediction and the (unexpected) omission of the aversive US during acquisition (based on the three differential contrasts reported above). Conjunction analyses revealed common areas of activation in the cerebellar hemispheres primarily on the left (local maximum Crus I; testing global null hypotheses at a threshold of p<0.05 FWE corrected level without TFC enhancement) (*Figure 4a*, see also *Supplementary file 1* Table 2). Additional common areas of cerebellar activation were present in the anterior and posterior vermis when considering the two contrasts related to US presentation and its unexpected omission only (*Figure 4—figure supplement 1b*).

Next, we were interested whether the level of activations differed between the three differential contrasts of interest. Using the same second level model, the main effect of contrasts was calculated (*F*-test, contrast vector: [1 -1 0; 0 1 -1]; p<0.05 FWE corrected without TFCE). Significant differences were found in the left lobule Crus I (with a small extension into lobule VI) and a small cluster in the anterior vermis (local maximum lobule I-IV) (*Figure 4b*, *Supplementary file 1* Table 3). This difference reflected a lower level of activation related to the prediction of the aversive stimulus compared to its presentation and unexpected omission (see small insert in *Figure 4b*). Comparing any two out of the three contrasts at a time, revealed no significant difference in the level of activations comparing the omission and the prediction of the aversive stimulus, except a small cluster in the anterior

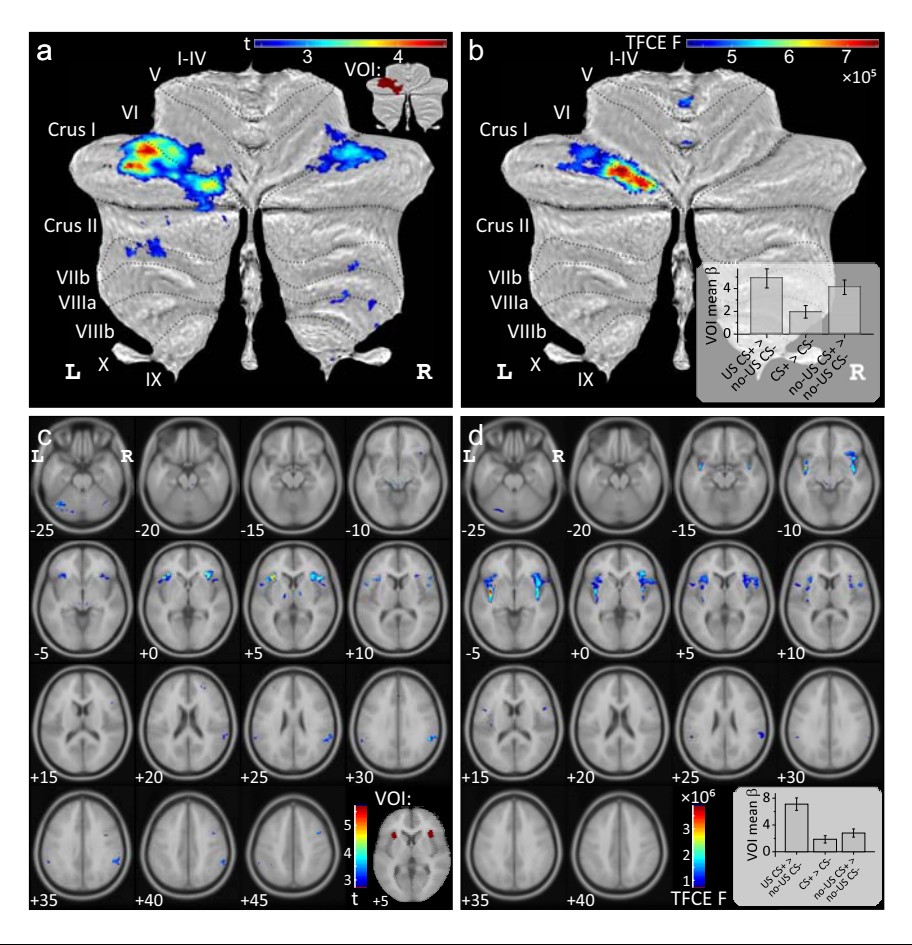

**Figure 4.** Conjunction analyses. Conjunction analyses testing global null hypotheses (**a,c**) and analyses of differences (**b,d**) between the three contrasts 'US post CS+ > no US post CS-', 'CS+ > CS-' and 'no-US post CS + > no US post CS-' (shown in *Figure 3*) during fear acquisition. Data in (**a,b**) is shown in SUIT space and in (**c,d**) in MNI space. All contrasts displayed using FWE correction (p<0.05), (**b,d**) using TFCE. Bar graphs display group mean $\beta$ values for each contrast considering the whole activation volume (error bars: standard error). VOI were defined based on conjunction analyses and are shown in the inserts: cerebellar VOI (**a**) and bilateral insula VOI (**c**). CS = conditioned stimulus; FWE = familywise error; TCFE = threshold-free cluster-enhancement; US = unconditioned stimulus; VOI = volumes of interest.

DOI: https://doi.org/10.7554/eLife.46831.010

The following source data and figure supplement are available for figure 4:

**Source data 1.** Mean $\beta$ data.

DOI: https://doi.org/10.7554/eLife.46831.012

**Figure supplement 1.** Comparison of cerebellar areas related to the presentation, the prediction and the omission of the aversive stimulus.

DOI: https://doi.org/10.7554/eLife.46831.011

---

vermis which was more prominent related to the omission of the US (*Figure 4—figure supplement 1e*). The level of activations of the vermis and neighboring areas of the cerebellar hemispheres were significantly higher related to the experience of the US compared to its prediction and unexpected omission (*Figure 4—figure supplement 1d,e*).

## Mean β values related to each event (presentation of US, CS+, CS-, omission of US) compared to rest

Based on the conjunction analyses for the three contrasts of interest in acquisition, we defined a volume of interest (VOI) in the left cerebellar hemisphere (indicated in red in the insert in *Figure 4a*). Mean β values were calculated for each event compared to rest within the VOI.

There was no significant effect of stimulus type (CS+ or CS-) during the habituation phase ($t_{21}$ = 0.86, p=0.397; *Figure 5a*). During acquisition, mean β values in CS+ trials (black triangles in *Figure 5a*) were significantly higher compared to CS- trials (inverted gray triangles; $F_{1,21}$ = 14.56, p=0.001). The block (early vs. late) effect ($F_{1,21}$ = 0.64, p=0.432) and stimulus type by block interaction ($F_{1,21}$ = 3.96, p=0.060) effects were not significant. During extinction, mean β values declined in CS+ trials, and were no longer different between CS+ and CS- trials ($F_{1,21}$ = 0.27, p=0.610). Block ($F_{1,21}$ = 3.94, p=0.060) and stimulus type by block interaction ($F_{1,21}$ < 0.01, p=0.973) effects were not significant.

Mean β values related to US events were significantly higher in response to the presentation of the aversive US in paired CS+ trials (US post CS+; black diamonds in *Figure 5a*) compared to the corresponding event in CS- trials (no-US post CS-; inverted black triangles) ($F_{1,21}$ = 26.75, p<0.001). There were no significant block (early vs. late; $F_{1,21}$ = 2.22, p=0.151) or stimulus type by block interaction effects ($F_{1,21}$ <0.01, p=0.960). Likewise, mean β values related to the unexpected omission of the US in CS+ trials (no-US post CS+; gray triangles) were significantly higher compared to the corresponding event in CS- trials in early ($t_{21}$ = 4.38, p<0.001) and late acquisition phases ($t_{21}$ = 6.73, p<0.001). During extinction, mean β values declined, but remained significantly higher related to no-US post CS+ events compared to no-US post CS- events ($F_{1,21}$ = 4.52, p=0.046). The block effect (early vs. late; $F_{1,21}$ = 8.326, p=0.009) was significant. The stimulus type by block interaction was not significant ($F_{1,21}$ = 0.08, p=0.784).

## PPI data: cerebello-cerebral interactions

As described above, a VOI was defined in the lateral cerebellar cortex based on conjunction analyses. Analyses of psychophysiological interactions (PPI) for the three contrasts of interest were performed between this cerebellar VOI and the whole brain. PPIs are reported which are significant at p<0.05 FWE corrected level after TFCE application. The most prominent finding was significant modulation of the functional connectivity between the cerebellum and occipital lobe during fear acquisition. There was no significant PPI found during extinction.

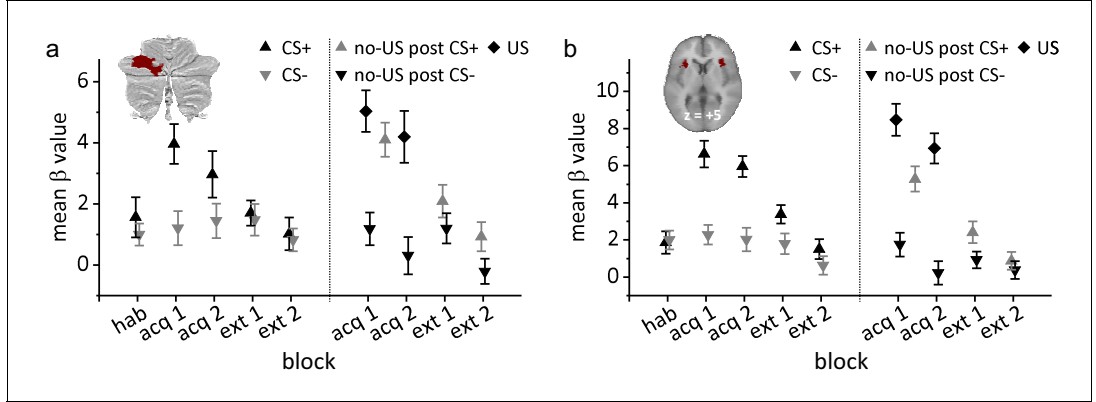

**Figure 5.** Group mean β values related to each event (presentation of US, CS+, CS-, omission of US) compared to rest. (a) Volume of interest (VOI) in the left cerebellar hemisphere; (b) VOI in the bilateral insula. Error bars represent standard errors. acq 1, acq 2 = early and late acquisition; CS = conditioned stimulus; ext 1, ext 2 = early and late extinction; hab = habituation; US = unconditioned stimulus.
DOI: https://doi.org/10.7554/eLife.46831.014

The following source data is available for figure 5:

**Source data 1.** Mean β data.
DOI: https://doi.org/10.7554/eLife.46831.015

### PPI related to the presentation of the aversive stimulus

Considering the seed region in the left cerebellar hemisphere, activation related to the presentation of the US (as revealed by the contrast "US post CS+ > no US post CS-") showed increased functional connectivity with striate and extrastriate visual areas (blue-green color code in *Figure 6*; see also *Table 2*; local maxima in the calcarine fissure and surrounding cortex (V1)). Additional areas of increased functional connectivity were found in limbic areas (cingulum, parahippocampus). No significant decreases of functional connectivity were found.

### PPI related to prediction of the aversive stimulus

Cerebellar activation in the hemispherical seed region related to prediction of the aversive stimulus (as revealed by the contrast 'CS+ > CS-') showed increased functional connectivity with extrastriate visual areas (local maxima in middle occipital lobe, lingual gyrus, fusiform gyrus; red-yellow color code in *Figure 6*; *Table 2*). No significant decreases of functional connectivity were found.

### PPI related to the (unexpected) omission of the aversive stimulus

Cerebellar activation in the hemispherical seed region related to the (unexpected) omission of the aversive US (as revealed by the contrast 'no-US post CS+ > no US post CS-') showed no significant increases or decreases of functional connectivity.

### 'Bleed over' from visual cortex

To exclude the possibility that the observed functional connectivity between cerebellum and occipital cortex was driven by 'bleed over' from visual cortex, data were reanalyzed using the mean signal modulation in the occipital lobe as first level nuisance regressor (see *Buckner et al., 2011*), for details). The main findings did not change (see *Figure 6—figure supplement 1*).

## Whole brain analysis

Although the focus of the study was on the cerebellum, exploratory whole brain fMRI analysis was also performed. Data are presented at p<0.05 FWE corrected level after TFCE application in all cases but conjunction analysis. Conjunction analysis results are reported at the level of p<0.05 FWE without TFCE. Most prominent activation was observed within the insula (*Figure 3*). Activation of other limbic areas was observed, primarily related to presentation of the aversive stimulus. Similar to cerebellar activation, prominent activation of the insula was observed not only to the prediction of the upcoming US but also related to its unexpected omission during acquisition trials. Cerebral activations vanished during extinction trials (at p<0.05 FWE corrected level after TFCE).

### Cerebral activation related to the presentation of the aversive stimulus [contrast 'US post CS+ > no US post CS-']

Most prominent activations were found in the insula bilaterally (*Figure 3d*). Additional differential activations were present in the anterior and middle cingulate gyrus, the amygdala, supplementary motor area (SMA), supramarginal and

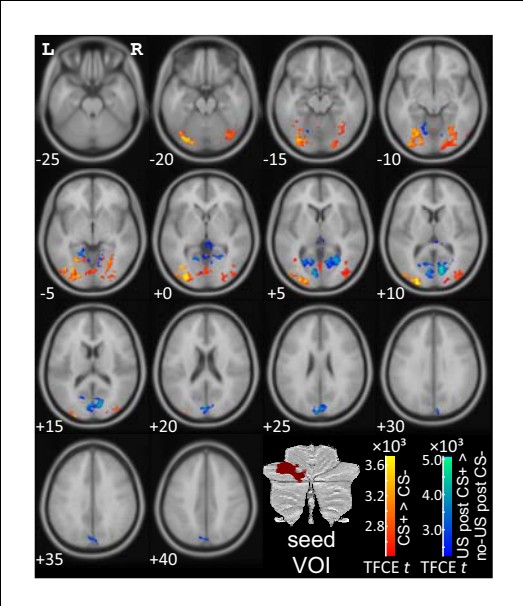

**Figure 6.** PPI analysis based on a seed region in the left lateral cerebellar cerebellum (p<0.05 FWE corrected level after TFCE application). CS = conditioned stimulus; FWE = familywise error; L = left; PPI = psychophysiological interaction; R = right; TFCE = threshold-free cluster-enhancement; US = unconditioned stimulus; VOI = volumes of interest.

DOI: https://doi.org/10.7554/eLife.46831.016

The following figure supplement is available for figure 6:

**Figure supplement 1.** Reanalysis of PPI data to exclude "bleed over" from visual cortex.

DOI: https://doi.org/10.7554/eLife.46831.017

**Table 2.** Psychophysiological interactions (PPI) based on a seed region in the left lateral cerebellum.
Clusters of ≥20 mm³ are shown. Up to three maxima in each cluster are shown separated by at least 8 mm.

| Index | Location | Side | SUIT coordinates/mm | | | Cluster size/ mm³ | $p_{FWE}$ | TFCE |
|---|---|---|---|---|---|---|---|---|
| *PPI (increased functional connectivity): acquisition, US post CS+ > no US post CS- t-test, TFCE, p<0.05 FWE corr.* | | | | | | | | |
| 1 | Extended cluster | right Calcarine (4607), left Calcarine (3955), left Cuneus (2921), left Lingual (1995), right Lingual (1903), outside GM (1632), right Cuneus (932), left Precuneus (424), vermal Lob. IV-V (323), left Lob. IV-V (193), left Lob. VI (141), left Occipital_Sup (117), right Thalamus (53), right Precuneus (47), left Thalamus (39), right Lob. IV-V (37), right Cingulum_Post (27), left Parietal_Sup (27), left Cingulum_Post (13), left ParaHippocampal (11), right Hippocampus (9), right ParaHippocampal (3) | | | | | | |
| | Calcarine | Right | 11 | −74 | 11 | 19409 | 0.001 | 5080.1 |
| | Calcarine | Left | -8 | −80 | 7 | | 0.002 | 4220.7 |
| | Calcarine | Right | 22 | −58 | 3 | | 0.003 | 4102.5 |
| *PPI (increased functional connectivity): acquisition, CS+>CSt-test, TFCE, p<0.05 FWE corr.* | | | | | | | | |
| 1 | Extended cluster | left Occipital_Mid (5908), left Lingual (4253), left Fusiform (2706), left Occipital_Inf (2297), outside GM (1491), left Lob. Crus I (645), right Lingual (616), left Calcarine (241), left Lob. VI (177), left Occipital_Sup (114), right Calcarine (99), left Precuneus (21), left Temporal_Mid (16), vermal Lob. IV-V (3) | | | | | | |
| | Occipital_Mid | Left | −24 | −98 | 10 | 18587 | 0.007 | 3714.2 |
| | Occipital_Mid | Left | −29 | −89 | 0 | | 0.008 | 3634.5 |
| | Fusiform | Left | −24 | −71 | -8 | | 0.009 | 3568.4 |
| 2 | Extended cluster | right Fusiform (3493), right Occipital_Mid (2820), right Lingual (2743), right Occipital_Inf (1068), outside GM (1040), right Temporal_Mid (411), right Occipital_Sup (329), right Cuneus (301), right Lob. Crus I (288), right Lob. VI (270), right Temporal_Inf (251), right Calcarine (151) | | | | | | |
| | Fusiform | Right | 33 | −79 | -7 | 13165 | 0.018 | 3160.4 |
| | Lingual | Right | 26 | −62 | -6 | | 0.018 | 3133.8 |
| | Fusiform | Right | 24 | −71 | -6 | | 0.02 | 3088.5 |
| 3 | Paracentral Lob. | Left | -2 | −38 | 69 | 394 | 0.018 | 3158.1 |
| 4 | Extended cluster | vermal Lob. IV-V (92), left Lob. IV-V (35), vermal Lob. VI (12), left Lob. VI (8) | | | | | | |
| | Lob. VI | Vermal | -1 | −64 | −10 | 147 | 0.04 | 2580.3 |
| | Lob. IV-V | Vermal | 1 | −56 | -4 | | 0.05 | 2417.9 |
| 5 | Extended cluster | left Fusiform (248), left Occipital_Inf (6), outside GM (1), left Temporal_Inf (1) | | | | | | |
| | Fusiform | Left | −33 | −48 | −13 | 256 | 0.042 | 2541.9 |
| | Fusiform | Left | −41 | −58 | −14 | | 0.042 | 2520.3 |
| 6 | Extended cluster | right Lingual (173), right Lob. IV-V (14) | | | | | | |
| | Lingual | Right | 17 | −56 | -4 | 187 | 0.044 | 2483.6 |
| | Lingual | Right | 10 | −60 | -4 | | 0.045 | 2475.6 |
| 7 | Lingual | Right | 17 | −52 | 2 | 65 | 0.047 | 2455.3 |

DOI: https://doi.org/10.7554/eLife.46831.013

superior temporal gyrus bilaterally, frontal inferior gyrus and cuneus (summarized in *Supplementary file 1* Table 5).

## Cerebral related to the prediction of the aversive stimulus [contrast 'CS + > CS-']

During acquisition, cerebral activation related to the CS+ was significantly higher compared to the CS- in the more anterior parts of the insula bilaterally (*Figure 3e*; *Supplementary file 1* Table 5). Additional differential activation was present in the right SMA and middle cingulate gyrus. During extinction, no significant fMRI activation was observed.

## Cerebral activation related to the omission of the aversive stimulus [contrast 'no-US post CS+ > no US post CS-']

During acquisition, significant differential activation related to the (unexpected) omission of the US was found in more anterior parts of the insula bilaterally (*Figure 3f*, *Supplementary file 1* Table 5). Additional activation was found bilaterally in the SMA, anterior and medial cingulate and right supra-marginal cortex. During extinction, no significant activations were observed.

## Comparison of cerebellar areas related to presentation, prediction and omission of the aversive stimulus

Conjunction analyses revealed that the more anterior parts of the insula bilaterally were activated in the three contrasts of interest (testing the global null hypotheses at a threshold of p<0.05 FWE corrected level without TFC enhancement) (*Figure 4c*, see also *Supplementary file 1* Table 2). In the anterior, but also posterior, parts of the insula, the level of activations differed between the three differential contrasts of interest (*F*-tests; p<0.05 FWE corrected without TFC enhancement; *Figure 4d*,*Supplementary file 1* Table 3). This difference reflected a higher level of activation related to the experience of the aversive stimulus compared to its prediction and unexpected omission (see small insert in *Figure 4d*).

## Mean β-values in the bilateral insula related to each event (presentation of US, CS+, CS-, omission of US) compared to rest

A VOI was defined in the bilateral insula based on conjunction analyses considering the three contrasts of interest as described above and a (bilateral) insula mask derived from the AAL template (see insert in *Figure 4c*).

There was no significant effect of stimulus type (CS+ or CS-) during the habituation phase ($t_{21}$ = −0.28, p=0.781; *Figure 5b*). During acquisition, mean $\beta$ values in CS+ trials (black triangles in *Figure 5b*) were significantly higher than in CS- trials (inverted gray triangles; $F_{1,21}$ = 22.13, p<0.001). The block effect (early vs. late; $F_{1,21}$ = 0.89, p=0.355) and stimulus type by block interaction ($F_{1,21}$ = 0.123, p=0.730) effects were not significant. During extinction, mean $\beta$ values in CS+ trials declined. The block effect was significant ($F_{1,21}$ = 7.58, p=0.012). Stimulus type ($F_{1,21}$ = 2.74, p=0.113) and stimulus type by block interaction effects ($F_{1,21}$ = 0.41, p=0.530) were not significant.

Mean $\beta$ values in the bilateral insula VOI were significantly higher in response to the presentation of the aversive US in paired CS+ trials (US post CS+; black diamonds in *Figure 5b*) compared to the corresponding event in CS- trials (no-US post CS-; inverted black triangles) ($F_{1,21}$ = 43.71, p<0.001). The block effect was significant (early vs. late; $F_{1,21}$ = 10.07, p=0.005). The stimulus type by block interaction was not significant ($F_{1,21}$ <0.01, p=0.991). Likewise, mean $\beta$ values related to the unexpected omission of the US in CS+ trials (no-US post CS+; gray triangles) were significantly higher compared to the CS- trials in early ($t_{21}$ = 4.53, p<0.001) and late acquisition ($t_{21}$ = 5.97, p<0.001). During extinction, mean $\beta$ values at the time of the presentation of the US in CS+ trials declined. Stimulus type effect was significant ($F_{1,21}$ = 4.49, p=0.046). Block ($F_{1,21}$ = 3.87, p=0.063) and stimulus type by block interaction ($F_{1,21}$ = 0.97, p=0.335) effects were not significant.

## Discussion

Cerebellar activation was observed related to the learned association of the CS and the aversive US confirming previous results (*Fischer et al., 2000*; *Frings et al., 2002*; *Ploghaus et al., 1999*). Most importantly, marked cerebellar activation was found also during the unexpected omission of the unpleasant event and disappeared during extinction trials (in which the omission became expected). These findings support the hypothesis that the cerebellum acts as or is part of a predictive device not only in the motor domain, but also in the emotional domain. In addition to the cerebellum, exploratory whole brain analysis showed very similar patterns of activation in the insula, which has been shown to be involved in aversive prediction error processing by others (*Geuter et al., 2017*; *Li et al., 2011*). Thus, first evidence was found that the cerebellum is part of a more extended neural network processing prediction errors in learned emotional responses. The discussion will focus on the cerebellar findings, which were also accompanied by changes in functional connectivity predominantly with visual cortices. Hence, one cerebellar role in emotional control may be to modulate processing of fear-related sensory information.

## Cerebellar activation during presentation and prediction of aversive events

Cerebellar activation during presentation and prediction of aversive events is in good accordance with the literature (*Dimitrova et al., 2003*; *Lange et al., 2015*; *Maschke et al., 2003*; *Ploghaus et al., 1999*). In a seminal study, *Ploghaus et al. (1999)* reported that distinct, but closely adjacent, cerebellar areas were related to the experience and the prediction of pain. In accordance, we found activation of the anterior cerebellum with a maximum in the cerebellar vermis related to presentation of the aversive stimulus. Different to *Ploghaus et al. (1999)*, however, cerebellar activation related to the experience of the US showed a significant extension to the posterolateral cerebellum (including lobules Crus I and VI) and overlapped with the area related to the prediction of the aversive stimulus. Furthermore, the level of activation in the posterolateral cerebellum was more related to the experience of the aversive stimulus compared to its prediction. Likewise, other fMRI studies have reported that aversive stimuli result in more widespread cerebellar activations of both anteromedial and posterolateral areas (painful electrical stimulation of the feet: *Dimitrova et al., 2003*; airpuffs directed to the eye: *Maschke et al., 2003*; *Moulton et al., 2010*, for review). Responses to aversive stimuli are complex and involve autonomic, sensorimotor, cognitive, and higher-order emotional reactions. Parts of the vermis are known to contribute to autonomic functions (*Apps et al., 2018*; *Apps and Strata, 2015*, for reviews) and emotion processing tasks (*Guell et al., 2018*, but see also *King et al., 2019*), the anterior lobe, part of lobule VI and lobule VIII to sensorimotor functions, and Crus I and II as well as part of lobule VI to cognitive functions (*Guell et al., 2018*; *King et al., 2019*; *Stoodley and Schmahmann, 2018*). Lobules VI, Crus II and X have also been related to emotional processes (*Guell et al., 2018*). Thus, different parts of the cerebellum likely contribute to the various aspects involved in processing of aversive stimuli (*Moulton et al., 2010*).

Very similar to our findings, a more recent fMRI study also reported an overlap of cerebellar areas related to the experience and prediction of painful stimuli in Crus I and lobule VI of the posterolateral cerebellum (*Michelle Welman et al., 2018*). Nevertheless, we suggest a different interpretation of the data from *Ploghaus et al. (1999)*: rather than reflecting a dissociation between the experience and the prediction of unpleasant events, different parts of the cerebellum are likely involved in the different aspects of learned fear. For example, midline parts of the cerebellum are likely involved in autonomic processes, and posterolateral parts of the cerebellum in cognitive and higher-order emotional processes related to the experience and prediction of potentially harmful stimuli. The known motor areas also likely contribute to this picture. Depending on stimulus intensity, participants may withdraw their hand or at least prepare a hand movement. In fact, early animal, but also human cerebellar lesion studies highlight the involvement of the cerebellar vermis in the conditioning of autonomic fear responses (*Apps and Strata, 2015*; *Apps et al., 2018* for reviews; *Sacchetti et al., 2002*; *Supple and Leaton, 1990a*; *Supple and Kapp, 1993*). For example, *Maschke et al. (2002)* found that fear-conditioned bradycardia was impaired in patients with lesions of the cerebellar midline but not the lateral cerebellar hemispheres. On the other hand, activation of the posterolateral cerebellar hemisphere is very common in human fear conditioning fMRI studies (*Lange et al., 2015*). Given the known reciprocal connections of the posterolateral cerebellum and its output nuclei, the dentate nuclei, with the prefrontal cortex (*Middleton and Strick, 1994*; *Middleton and Strick, 2000*), lateral activations may reflect the more cognitive aspects of fear conditioning. Contingency awareness may play a critical role: a prerequisite for conditioned fear responses to occur is that participants become aware of the CS-US contingencies (*Dawson and Furedy, 1976*; *Lonsdorf et al., 2017*; for reviews; *Tabbert et al., 2006*; *Tabbert et al., 2011*). Contingency awareness is likely linked to working memory processes (*Dawson and Furedy, 1976*). In fact, cerebellar activation in lobules VI and Crus I overlaps with areas in the cerebellum which have been shown to contribute to working memory processes (e.g. *Guell et al., 2018*; *King et al., 2019*, although in their study 'active maintenance' was lateralized to the right; and visual working memory was localized in the vermis). If this assumption is true, the pattern of cerebellar activation should be different in aware and unaware participants. It will be of high interest to test this hypothesis in future studies, for example using multiple CSs (*Rehbein et al., 2014*) or using distractor tasks (e.g. *Tabbert et al., 2006*; *Tabbert et al., 2011*) to prevent awareness of the CS-US contingencies.

In the present study, the focus of activation was within Crus I with some extension into lobule VI. Although it cannot be excluded that part of the activation is related to the preparation or subliminal execution of a withdrawal movement, motor-related processes are unlikely to explain the bulk of posterolateral activation. Hand and finger movements result in fMRI activation of ipsilateral lobule V, with additional activation of lobule VI bilaterally in more complex movements (*King et al., 2019*; *Schlerf et al., 2010*). Although some extension to Crus I has been observed in the latter, movements never result in activations primarily of Crus I. Rather, focus of activation is always on lobules V and VI, which was clearly not the case in the present study. Of note, preparation and execution of movements have been found to activate the same cerebellar areas (*Cui et al., 2000*).

The present findings agree with the classic view that prediction depends on activity in the same networks that process the actual experience (e.g. *James, 1892*), at least at the level of the cerebellum. Recent single-cell recording studies within the cerebellar cortex in monkeys are also in line with this assumption: both simple and complex spike firing rates at the same Purkinje cell encode movement kinematics and sensory feedback, but also motor predictions (*Popa et al., 2012*; *Streng et al., 2017a*; *Streng et al., 2017b*).

Based on animal and human lesion data, vermal activation is to be expected related to the prediction of the aversive stimulus. A recent fMRI meta-analysis indeed showed activations of both the cerebellar hemispheres and the vermis in fear conditioning paradigms in healthy humans (*Lange et al., 2015*). This was, however, neither the case in the present study nor in the study by *Ploghaus et al. (1999)*. Because participants were instructed about the CS-US contingencies to a certain degree in both studies, the cognitive component may have had the strongest impact on the fMRI data.

*Ploghaus et al. (1999)* reported a similar dissociation for the experience and the prediction of pain in posterior and anterior parts of the insula, respectively. In the present study, we also found activation related to the US in the posterior insula, and activation related to the CS+ (and therefore prediction of the US) in more anterior parts. Very similar to our cerebellar findings, however, US-related activation extended into the more anterior insula and overlapped with CS+ related activations. Again, we hypothesize that this is not a dissociation between experience and prediction but reflects different functional aspects of processing of potentially harmful stimuli. For example, *Frot et al. (2014)* suggested that posterior parts of the insula may be more important in the evaluation of the intensity and localization of an aversive stimulus, whereas the anterior insula may process the emotional reaction to the stimulus. However, as yet, the anterior insula, but not the posterior insula has been shown to be involved in processing predictions of aversive events (*Geuter et al., 2017*). Notwithstanding, this concept may also be extended to other brain areas, for example. the medial and anterior cingulate cortex (*Vogt, 2014*). In the present study, however, no other cerebral regions showed significant activations related to the prediction of the aversive stimulus in the whole brain analysis using a conservative statistical threshold.

It should be noted that a nonselective analysis of cerebellar (and cerebral) activations has been performed, that is interpretation of specific regional activations has been done a posteriori. Another approach would comprise a selective analysis of (a priori) functionally defined ROIs based on the recent studies mapping the cerebellum to a broad range of functions (*Guell et al., 2018*; *King et al., 2019*). As yet, however, there are important open questions regarding cerebellar areas involved in emotions. The emotional area described by *Guell et al. (2018)* is based on one single task, not on a meta-analysis of a significant number of different emotional tasks. In the study by *King et al. (2019)*, more tasks included emotional processing. However, emotional processing did not become part of any of the 10 main functional regions of the cerebellum introduced by the authors. Furthermore, they found that vermal activation was best explained by eye movements (or visual attention), but not emotional processing as reported by *Guell et al. (2018)*. Thus, as yet, a nonselective analysis gives a more unbiased picture of cerebellar regions involved in fear conditioning. We do acknowledge, however, that this approach allows only limited conclusions on the possible contributions of selective cerebellar areas to learned fear responses.

## Cerebellar activation during the unexpected omission of predicted aversive events

We found cerebellar activation related to the predicted occurrence of an aversive US. More prominent cerebellar activations, however, were observed during the unexpected omission of the

unpleasant event. Cerebellar activation was most marked in the posterolateral cerebellum (lobules Crus I, VI), but additional activations were also present in the vermis. Importantly, cerebellar activation vanished during extinction trials, during which the omission of the US became expected. These findings support the hypothesis that the cerebellum is involved in encoding and/or processing of prediction errors. The present findings are supported by earlier findings (*Ploghaus et al., 2000*) reporting activations of the posterolateral cerebellar hemisphere in the very first extinction trial in an associative learning task using painful heat stimuli as US. Our findings are also very similar to findings in a recent fMRI study on the cerebellar contributions to language (*Moberget et al., 2014*): Activation of the posterolateral cerebellum (Crus I and II) was related to the predictability of upcoming words in a sentence (e.g., two plus two is *four*). Similar to the present findings, prominent cerebellar activation was also observed when this prediction was violated (e.g., two plus two is *apple*). In the sensorimotor domain, fMRI data in humans also show cerebellar activation related to the unexpected omission of an expected sensory stimulus (*Ramnani et al., 2000*; *Schlerf et al., 2012*).

Based on theoretical models it has long been assumed that error information is sent to the cerebellar cortex via the climbing fibers (*Albus, 1971*; *Marr, 1969*). Climbing fibers have been shown to signal the unexpected occurrence and the unexpected omission of the airpuff-US in eyeblink conditioning in mice and rabbits (see *Ohmae and Medina, 2015*, for a recent study): Whereas the unexpected occurrence leads to an increase of climbing fiber activity, the unexpected omission results in a decrease. Because the fMRI signal is thought to reflect synaptic activity (*Lauritzen et al., 2012*), decrease of climbing fiber input cannot explain the observed increased fMRI signal in the cerebellar cortex during US omission. Prediction errors, however, may not only be signaled by the climbing fiber system. There is also evidence that mossy fibers play a role (*Popa et al., 2017*; *Streng et al., 2018*). Furthermore, the role of the cerebellum may go beyond the processing of sensory predictions and sensory prediction errors and may include reward predictions and prediction errors (*Carta et al., 2019*; *Wagner et al., 2017*). *Wagner et al. (2017)* found granule cells that responded preferentially to reward, to reward omission and reward anticipation, a function commonly ascribed to the dopaminergic system (*Schultz et al., 1997*; *Schultz, 2017*). Importantly, reward omission granule cells were significantly more frequent than reward cells. In addition to sensory and reward prediction errors, the cerebellum may be involved in prediction of punishment and punishment prediction errors. As outlined in the introduction, several brain areas are likely involved in the processing of predictions and prediction errors in associative fear learning. As yet, it is unknown where prediction errors of learned fear responses are encoded (*Tovote et al., 2015*). Because there is some experimental evidence that sensory prediction errors are encoded in the cerebellar cortex and nuclei (*Brooks et al., 2015*; *Ohmae and Medina, 2015*; *Popa and Ebner, 2018*), the cerebellum is a likely candidate. However, this issue is far from being settled and extracerebellar areas may also play a role.

## Changes in connectivity between the cerebellum and visual cortex

Functional connectivity of the cerebellum was increased with visual cortical areas when comparing CS+ with CS- trials and with limbic areas during presentation of the US, but not during its prediction. Likewise, *Lithari et al. (2016)* found that visual cortex processing plays a more central role and that limbic areas become functionally decoupled in a fear conditioning paradigm. Increased connectivity between the cerebellum and visual cortex suggests that the cerebellum contributes to the known enhancement of the perception of visual stimuli during fear conditioning (*Petro et al., 2017*).

However, at first sight, increased connectivity between the cerebellum and visual cortex is unexpected. In monkeys, the primary visual cortex has no known afferent connections with the cerebellum (*Glickstein et al., 1994*; *Schmahmann and Pandya, 1997*). Likewise, resting state fMRI revealed no functional connectivity between the cerebellum and primary visual cortex in a large study population (*Buckner et al., 2011*). Rather, the cerebellum receives dense afferent connections from the dorsal stream of parietal lobe visual areas (*Glickstein, 2000*; *Schmahmann and Pandya, 1997*) and is known to increase visual perception of movements (e.g., *Christensen et al., 2014*; *Händel et al., 2009*). The influence of the cerebellum on the perception of fear-conditioned visual stimuli may be indirect: enhanced processing of fear conditioned visual stimuli in the visual cortex has been shown to be under the control of cortical structures, in particular the middle frontal gyrus (MFG; *Petro et al., 2017*). Bidirectional cerebello-frontal connections are known for the frontal eye field and the dorsolateral prefrontal cortex, which play an important role in attention (*Middleton and*

*Strick, 2001*). Possibly, the cerebellum may help to increase selective attention to the CS. The present findings, however, are based on PPI, which do not give any information about directionality.

## Conclusions

The most important present finding is the pronounced cerebellar activation during the unexpected omission of a predicted aversive stimulus. This cerebellar activation is best explained by the generation or further processing of prediction errors. As expected, cerebellar activation was also found during the prediction of aversive stimuli. These findings support the hypothesis that the cerebellum is of general importance for predictive control including the emotional domain. The cerebellum has to be added to the more extended neural network involved in processing of aversive predictions and prediction errors.

# Materials and methods

## Participants

Experiment power was estimated based on previous eyeblink conditioning data, which had also been acquired at 7 T (*Ernst et al., 2017*) using the fmriPower toolbox for MATLAB (fmripower.org; *Mumford, 2012*). Considering CS-only trials in acquisition and aiming for a power of 80% at p<0.001, group sizes were estimated to 21 participants for lobule VI and 30 participants for Crus I ipsilaterally to US application.

A total of 27 young and healthy participants performed the experiment. Three participants had to be excluded because of technical errors, one participant because of an incidental finding on brain MRI, and one participant because of constant motion throughout MRI acquisition. Thus, a total of 22 participants (eight males, 14 females, mean age: 26.9 (SD = 4.3) years, range: 19–32 years) were included in the final data analysis. None of the participants presented with neurological or neuropsychiatric disorders based on medical history. None were taking centrally acting drugs, except two who were taking a low dosage of a corticosteroid and an antihistamine, respectively. All participants were right-handed based on the Edinburgh handedness inventory (*Oldfield, 1971*) and had normal or corrected-to-normal vision. They were asked to refrain from alcohol consumption the night before the experiment. Informed consent was obtained from all participants. The study was approved by the local ethics committee and conducted in accordance with the Declaration of Helsinki.

## Fear conditioning

The entire experiment was performed within one session inside the MRI scanner. The paradigm presentation was controlled by a computer running the software Presentation (version 16.4, Neurobehavioral System Inc, Berkeley, CA). *Figure 1* displays the experimental paradigm. Participants were shown images of the visual stimuli used in the experiment and told that electrical shocks would be applied during the experiment. They were instructed that, should they perceive a pattern between CS and US presentations, the experimenter would not change it during the experiment.

Visual stimuli were projected onto a rear projection screen inside the scanner bore using a standard projector. Images were visible to the participants through a mirror mounted on the radiofrequency (RF) head coil. Two pictures of black-and-white geometric figures (a square and a diamond shape, that is the square tilted by 45°) of identical brightness were used as CS+ and CS- (time of presentation: 8 s). In reinforced CS+ trials (i.e. 10 out of 16 acquisition trials), the visual stimulus co-terminated with the presentation of the aversive US. In CS- trials, the visual stimulus was never followed by the aversive US. A neutral black background image was displayed in between visual stimulus presentations (ITI randomized between 16 s and 20 s). Use of the two figures as CS+ and CS- was pseudorandomly counterbalanced between the individual participants.

A short electrical stimulation was used as an aversive US. The electrical stimulation was generated by a constant current stimulator (DS7A, Digitimer Ltd., London, UK) and applied to the left hand via a concentric (ring-shaped) bipolar surface electrode with 6 mm conductive diameter and a central platinum pin (WASP electrode, Specialty Developments, Bexley, UK). For MR-safety reasons, an in-house build non-magnetic high-resistivity electrode lead was used to connect the stimulator with the surface electrode (*Schmidt et al., 2016*). The 100 ms US consisted of a short train of four consecutive 500 µs current pulses (maximum output voltage: 400 V) with an inter pulse interval of 33 ms.

Immediately before the start of MRI measurements, the stimulation current was gradually increased, and participants were asked to report on the perceived sensation intensity until an 'unpleasant but not painful' intensity was reached (mean current: 3.9 (SD = 2.3) mA, range 1.6–9.3 mA). The final individual current setting was kept constant for all stimulations. Stimulus timing was set for the US to co-terminate with visual CS+ presentation.

During the experiment three types of trials were presented to the participants: CS+ followed by an US (paired CS+/US trial), CS+ without an US (CS+ only trial) and CS- without US (CS- only trial). The experimental protocol consisted of the three phases: 'habituation' (four CS+ only trials, four CS- only trials, presented in alternating order), 'acquisition' (10 paired CS+/US trials, six CS+ only trials, 16 CS- only trials) and 'extinction' (16 CS+ trials, 16 CS- only trials). Different trial types in acquisition and extinction were presented in a pseudorandomized order with four restrictions: firstly, the first two trials of acquisition were set to be paired CS+/US trials; secondly, there were never more than two consecutive CS of one kind presented in a row; thirdly, during acquisition and extinction the number of events of each kind was kept identical in the first half and in the second half of the experiment; and fourthly, the very last trial of acquisition was set to be a paired CS+/US trial. During acquisition, the order of events was the same for all participants, while use of the two different figures as CS+ and CS- was counterbalanced across the whole group. Order of CS+ and CS- events was counterbalanced during extinction (12 starting with CS+, 10 starting with CS-), and habituation (15 starting with CS+, seven starting with CS-). Each experimental phase was performed within a separate block of fMRI data acquisition.

Of note, fear conditioning in both animals and humans are commonly done with long CS-US intervals because measures of autonomic fear responses evolve slowly. Different to eyeblink conditioning, fear conditioning is tolerant to these long CS-US periods (*Carter et al., 2003*; *Lonsdorf et al., 2017*). This allows performance of event-related fMRI studies, which can separate between CS-related activation, US-related activation, and activation related to the omission of the US. Both animal (*Lavond et al., 1984*; *Sacchetti et al., 2002*; *Supple and Leaton, 1990a*; *Supple and Leaton, 1990b*; *Supple and Kapp, 1993*) and human studies (*Maschke et al., 2002*) have shown that the cerebellar vermis is involved in fear conditioning – despite these long CS-US time intervals. The contribution of the cerebellum to discrete somatic motor behavior (such as eyeblink responses) may be different compared to slowly reacting autonomic responses.

## Questionnaires

Participants were required to answer three questionnaires, one before the start of the experiment, a second one in between acquisition phase and extinction phase, and a third questionnaire after the experiment. The first and the third questionnaires were print copies handed out to the participant. The second questionnaire was projected onto the screen inside the MRI scanner bore one question at a time and answers were given orally via an intercom system.

Participants were asked to rate their (hedonic) valence and (emotional) arousal on viewing images of the CS+ and CS- on a nine-step Likert scale from 'very unpleasant' to 'very pleasant' and 'quiet and relaxed' to 'very excited', respectively. Additionally, the questionnaire following acquisition contained five questions regarding US perception and CS-US contingency: rating of the last US on a nine step-scale ('not unpleasant' to 'very unpleasant'); a multiple-choice question and a percentage estimate whether the US was applied after the presentation of the square and the diamond-shaped CS (options: 'always', 'sometimes', 'never', 'I cannot answer'); and lastly an estimation after which time and number of US presentations, if at all, a connection between the visual stimuli and the US presentation was identified.

Statistical analyses were performed using SPSS software (Version 24, RRID:SCR_002865, IBM Corp., Armonk, NY). Using repeated measure analyses of variance (ANOVA) valence and arousal ratings were tested for within-subject effects of stimulus type (CS+ vs. CS-) and phase (pre-acquisition, post-acquisition vs. post-extinction). Where necessary, individual ratings were compared with post hoc *t*-tests.

## Physiological data acquisition

Physiological data measured throughout the experiment were SCR, pulse rate and breathing rate. SC was acquired using a physiological data acquisition station with a dedicated MRI-compatible SC

module and appropriate hardware filters sampling at 2 kHz (EDA 100C-MRI, BIOPAC Systems Inc, Goleta, CA). SC electrodes were attached to the participants' left middle and ring fingers.

Pulse rate and breathing rate were measured using the physiologic monitoring unit (PMU) provided by the MRI scanner (Siemens Healthcare GmbH, Erlangen, Germany) at a fixed sampling rate of 50 Hz. In detail, pulse oximetry signals were recorded using a wireless recording device clipped to the participant's right index finger. A respiratory bellows was attached to the participant's lower abdomen using a hook-and-loop belt. Of note, pulse and breathing data were used to perform physiological denoising of MRI data only (*Glover et al., 2000*). The low sampling rate as well as the broad peaks of the pulse curves did not allow for the accuracy needed to observe small peak-to-peak heart rate variations related to fear conditioning.

## Skin conductance analysis

To eliminate high-frequency noise and low-frequency drifts, SC data were bandpass filtered (−61 dB Blackman FIR filter, 0.5 to 10 Hz) using AcqKnowledge software (BIOPAC Systems Inc, Goleta, CA). All further SC data processing was performed using MATLAB software (Release 2017a, RRID:SCR_001622, The MathWorks Inc, Natick, MA). Semi-automated peak detection was performed, and SCR were defined as the maximum trough-to-peak-amplitude of any SCR peak within a given time interval. In each trial, SCR were evaluated for three distinct time windows (*Prokasy and Ebel, 1967*): the FIR within a time window of 1.0 s to 5.0 s after CS onset, the SIR within a time window of 5.0 s to 8.5 s after CS onset, and the unconditioned response window (i.e. TIR) 8.5–13.0 s after CS onset (irrespective of whether a US was presented in the particular trial or not) (*Figure 2b*). To normalize data, SCR values were increased by 1 µS and logarithmized (*Boucsein, 2012*; *Venables and Christie, 1980*). Mean SCR values were calculated grouped for blocks of four, five, six and eight events, corresponding to the first level regressor selection in MRI analysis.

Statistical analyses were performed using SPSS software (Version 24, IBM Corp., Armonk, NY). ANOVA with repeated measures were calculated for within subject effects of stimulus type (CS+ vs. CS-) and block (early vs. late) considering SCR values as dependent measure. Appropriate post hoc *t*-tests were calculated. Because there was only one block of unpaired CS+ trials in acquisition (see *Figure 1*), differences of TIR in unpaired CS+ trials with TIR in paired CS+ and CS- trials were analyzed using *t*-tests.

## MRI acquisition

All MR images were acquired with the participants lying supine inside a whole-body MRI system operating at 7 Tesla magnetic field strength (MAGNETOM 7T, Siemens Healthcare GmbH, Erlangen, Germany) equipped with a one-channel transmit/32 channel receive RF head coil (Nova Medical, Wilmington, MA). To homogenize the RF excitation field (B1), three dielectric pads filled with high-permittivity fluid were placed below and on either side of each participant's upper neck (*Teeuwisse et al., 2012*). As needed, further cushions were used to fix the head position within the RF coil.

Prior to fMRI acquisition a sagittal MP2RAGE sequence (*Gallichan and Marques, 2017*; *Marques et al., 2010*) was run to acquire whole-brain anatomical reference images with an isotropic voxel size of 0.75 mm. Further imaging parameters were set as follows: TR/TE, 6000/3.45 ms, TI1/TI2, 800/2700 ms, flip angles 1/2, 4°/5°, parallel acceleration factor, 3, phase and slice partial Fourier factor, 6/8, acquisition matrix, 320 × 300, number of slices, 192, TA, 9:40 min.

Whole brain functional fMRI acquisition was performed using a fat-saturated, two-dimensional simultaneous multi slice echo planar image (SMS-EPI) sequence (*Cauley et al., 2014*; *Setsompop et al., 2012*) with an isotropic voxel size of 1.7 mm, in three consecutive episodes for habituation (90 volumes), acquisition and extinction (320 volumes each). Imaging parameters were selected as follows: TR/TE, 2000/22 ms, flip angle, 70°, parallel acceleration factor, 2, SMS factor, 3, phase partial Fourier factor, 6/8, acquisition matrix, 130 × 130, number of slices, 90.

## Image processing

All image and fMRI analyses were performed using SPM 12 (RRID:SCR_007037, Wellcome Department of Cognitive Neurology, London, UK) on a platform running MATLAB on Mac OS X 10.12.6, if

not explicitly stated otherwise. SPM default brightness threshold was set from 0.8 to 0.1 to avoid signal dropouts within the hypointense cerebellar nuclei (*Thürling et al., 2015*).

Functional MRI volumes for each participant were corrected for slice timing and realigned to the first volume of the habituation phase. One mean functional volume per participant was calculated.

Brain extraction was performed on non-denoised uniform T1 weighted (UNI) volumes using the CBS tools for high-resolution processing of high-field brain MRI (*Bazin et al., 2014*). Best coregistration of the mean functional volume to the brain extracted structural volume was achieved using the function 'epi_reg' available in FSL (Release 5.0.10, RRID:SCR_002823, Centre for Functional MRI of the Brain, Oxford, UK). Coregistration was subsequently applied to all functional volumes of each respective participant.

Normalization of the cerebellum was performed using the SUIT-toolbox for SPM (version 3.1, RRID:SCR_004969). Using the spatially unbiased atlas template of the human cerebellum (*Diedrichsen, 2006*), brain-extracted structural volumes were segmented and cerebellar masks were generated. Manual correction of each mask was performed by an experienced technician using MRIcron software (RRID:SCR_002403, *Rorden and Brett, 2000*) and slightly smoothed with an isotropic smoothing kernel of 1.5 mm (and subsequently thresholded against a value of 0.5) to remove remnant single outlying voxels. Using this process, on average 6.2% (SD 2.0%) of voxels (predominantly in the anterior region) were removed from, and 3.6% (SD 1.3%) of voxels (mainly in the dorsal region) were added to the cerebellar masks. The segmented structural images and the cortical cerebellar mask were supplied to a region of interest (ROI)-based DARTEL normalization algorithm available within the SUIT toolbox, and a cerebellar normalization was calculated. In addition, whole-brain normalization to MNI-space was obtained using the SPM segment routine. Functional volumes were then separately normalized to SUIT space using the corrected cerebellar mask from SUIT segmentation and MNI space (whole brain). All normalized volumes were subsequently smoothed by an isotropic smoothing kernel of 4 mm.

## fMRI analysis

We focused our fMRI analysis on the cerebellum. In addition, exploratory analysis of the whole brain was performed. The first level analysis was modelled as an event related-design. All US, no-US and CS event regressors were modelled for the respective stimulus onset and all the respective durations were set to 0 s. The first five volumes of each fMRI run were disregarded. Events were blocked into 19 regressors of interest as displayed in *Figure 1*. If number of trials allowed (n ≥ 4), events of each kind were grouped in two equal-sized blocks representing the first (early) and the second (late) half of each phase. Regressors were chosen for CS+ and CS- during habituation (two regressors, four events each), CS+ and CS- presentations during acquisition and extinction (eight regressors, eight events each), US presentations during acquisition (two regressors, five events each), and the omission of US presentations (no-US) at the expected time of US presentations after CS onset (no-US post CS+: one regressor, six events during acquisition, two regressors, eight events each during extinction; no-US post CS-: four regressors, eight events each during acquisition and extinction).

To correct for motion, volume realignment parameters were prepared as six nuisance regressors (three translations and three rotations). Pulse oximetry and respiration data from PMU were processed using essential features of the PhLEM toolbox for SPM (*Verstynen and Deshpande, 2011*). To correct for physiological motion effects the RETROICOR (retrospective image-based correction) method was applied and eight regressors were generated (*Glover et al., 2000*), resulting in a total of 14 nuisance regressors for each fMRI run. To ensure that no significant information was lost as a result of physiological denoising, an additional analysis with motion regressors but without RETROICOR was performed. Similarly, to ensure that no fine details were lost as a result of smoothing, a further analysis was performed using the normalized, un-smoothed functional data.

First level main effect contrasts against rest and differential first level contrasts were generated and tested in second level t-tests. The contrast 'US post CS+ > no US post CS-' was calculated to reveal activation in response to the presentation of the aversive stimulus (US). The contrast 'CS + > CS-' was calculated to reveal activation related to the prediction of the US. Finally, the contrast 'no-US post CS+>no US post CS-' was calculated to reveal activation related to the omission of the US. For second level t-tests, first level differential contrasts for acquisition and extinction phase were collapsed over early and late blocks, respectively. Contrasts were never collapsed across experimental phases. To evaluate differences between early and late acquisition and extinction, individual

second level within-subject ANOVA were modelled for the contrasts 'CS+ > CS-' and "no-US post CS+ > no US post CS-', and (for acquisition only) for the contrast 'US post CS+ > no US post CS-'. TFCE was applied using the TFCE toolbox for SPM12 (R164 and R174, http://dbm.neuro.uni-jena.de/tfce/).

In addition, second level one-way ANOVA was modeled for the three main acquisition contrasts (considering early and late acquisition together). To identify regions of shared activation, conjunction analysis (*Price and Friston, 1997*) was performed to test global null hypotheses for each of the three contrasts. Using the same second level model, the main effect of contrast was calculated to assess differences between the three contrasts (*F*-test, contrast vector: [1 -1 0; 0 1 -1]).

PPI were modelled for the whole brain analysis (*Friston et al., 1997*) using cerebellar VOI based on conjunction analysis in SUIT space as seed regions. TFCE was applied on the results. Additionally, region mean $\beta$ values for selected VOIs were extracted from simple first level $\beta$ maps against rest and compared between CS and US events using ANOVA and paired *t*-tests. To exclude the possibility that the observed functional connectivity between cerebellum and occipital cortex was driven by 'bleed over' from visual cortex, PPI was reanalyzed using the mean signal modulation in the section of occipital lobe closer than 6 mm to the cerebellum as first level nuisance regressor, in analogy to the work performed by *Buckner et al. (2011)*.

Activation maps were masked in SUIT space using the SUIT atlas volume (Cerebellum-SUIT.nii) with the inner-cerebellar white matter manually filled in, and in MNI space using the SPM canonical inner-cranial volume mask (mask_ICV.nii). To display results, cerebellar (SUIT space) activation maps were projected on cerebellar flatmaps (*Diedrichsen and Zotow, 2015*). Whole brain MNI space activation maps were projected on MNI152 average T1 volume provided with SPM (icbm_avg_152_t1_-tal_lin.nii). To acquire anatomical region labels, maps were then projected onto the SUIT atlas volume (Cerebellum-SUIT.nii, *Diedrichsen, 2006*) and the automated anatomical labeling (AAL) atlas volume (AAL.nii, RRID:SCR_003550, *Tzourio-Mazoyer et al., 2002*), respectively.

## Acknowledgements

The authors would like to thank M Craske for her valuable advice and fruitful discussions, J Marquez for his support using the MP2RAGE sequence, B Poser for his work on the SMS-EPI sequence, and T Otto for his work on SCR analysis. This work was supported by a grant from the German Research Foundation (DFG; project number 316803389 – SFB 1280) to DT and HHQ (subproject A05), CJM (subproject A09), and UB (subproject A11).

## Additional information

### Funding

| Funder | Grant reference number | Author |
| --- | --- | --- |
| Deutsche Forschungsge-meinschaft | 316803389 (SFB1280, project A11) | Ulrike Bingel |
| Deutsche Forschungsge-meinschaft | 316803389 (SFB1280, project A05) | Harald H Quick Dagmar Timmann |
| Deutsche Forschungsge-meinschaft | 316803389 (SFB1280, project A09) | Christian Josef Merz |

The funders had no role in study design, data collection and interpretation, or the decision to submit the work for publication.

### Author contributions

Thomas Michael Ernst, Conceptualization, Data curation, Software, Formal analysis, Validation, Investigation, Visualization, Methodology, Writing—original draft, Project administration, Writing—review and editing; Anna Evelina Brol, Conceptualization, Data curation, Formal analysis, Validation, Investigation, Writing—review and editing; Marcel Gratz, Christoph Ritter, Conceptualization, Resources, Methodology, Writing—review and editing; Ulrike Bingel, Conceptualization, Methodology, Writing—review and editing; Marc Schlamann, Resources, Data curation,

Investigation, Writing—review and editing; Stefan Maderwald, Conceptualization, Resources, Investigation, Writing—review and editing; Harald H Quick, Conceptualization, Resources, Funding acquisition, Project administration, Writing—review and editing; Christian Josef Merz, Conceptualization, Resources, Software, Formal analysis, Supervision, Methodology, Writing—original draft, Project administration, Writing—review and editing; Dagmar Timmann, Conceptualization, Resources, Supervision, Funding acquisition, Methodology, Writing—original draft, Project administration, Writing—review and editing

### Author ORCIDs
Thomas Michael Ernst (iD) https://orcid.org/0000-0002-2170-9241

### Ethics
Human subjects: The study was approved by the ethics committee of the Medical Faculty of the University of Duisburg-Essen (16-7225-BO) and conducted in compliance with the WMA Declaration of Helsinki. All participants gave their informed oral and written consent, including their consent to publish.

### Decision letter and Author response
Decision letter https://doi.org/10.7554/eLife.46831.023
Author response https://doi.org/10.7554/eLife.46831.024

## Additional files

### Supplementary files
• Supplementary file 1. Five supplementary tables are supplied to report on further statistical results. Table 1, first interval skin conductance responses; Table 2, changes in cerebellar activation across blocks during acquisition and extinction; Table 3, results of cerebellar and whole brain conjunction analyses; Table 4, differences across acquisition contrasts in cerebellar and whole brain activation; Table 5, whole brain activations during acquisition and extinction.
DOI: https://doi.org/10.7554/eLife.46831.018
• Transparent reporting form
DOI: https://doi.org/10.7554/eLife.46831.019

### Data availability
Group level statistical results (Figure 3, 4, and 6) were uploaded to NeuroVault.org. The consent form that participants signed does not allow us to share the raw data publicly but it can be made available on request to interested researchers through a data sharing agreement.

The following dataset was generated:

| Author(s) | Year | Dataset title | Dataset URL | Database and Identifier |
|---|---|---|---|---|
| Ernst TM, Brol AE, Timmann D | 2019 | Differential Fear Conditioning Paradigm at 7T with Cerebellar Focus | https://neurovault.org/collections/OMLBHQAA/ | NeuroVault, OMLBHQAA |

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
