## [Decision Letter]

Thank you for submitting your article "The cerebellum is involved in processing of predictions and prediction errors in a fear conditioning paradigm" for consideration by *eLife*. Your article has been reviewed by two peer reviewers, and the evaluation has been overseen by Sam McDougle as the Reviewing Editor and Richard Ivry as the Senior Editor. The following individuals involved in review of your submission have agreed to reveal their identity: Torgeir Moberget and Maedbh King.

Your submission has been favorably evaluated though several major concerns must be addressed for re-submission of the manuscript. We have discussed the reviews with one another and the Reviewing Editor has drafted this decision to help you prepare a revision.

Summary:

The authors present an fMRI fear conditioning paradigm, and test the prediction that acquired responses to visual conditioning stimuli (i.e., shape images) and a paired aversive tactile unconditioned stimulus (i.e., a brief electric shock) will be observed in the cerebellum during the conditioning process, and will change predictably over the course of learning and extinction. The fMRI results appear to support these predictions, showing reliable activity in lateral regions of the cerebellum in response to the conditioning stimulus (CS+) versus a non-paired stimulus (CS-), and also in response to an omitted US following a CS+, subsequent to the acquisition of the fear association. Moreover, PPI analyses reveal learning-related connectivity patterns between the cerebellum and striate visual cortex, as well as the insula, implicating the cerebellum in a wider network of regions contributing to the conditioning process. These results in the affective domain support the broader idea that the cerebellum is a generalized predictive substrate, one that is involved in various learning domains outside of the traditionally highlighted role in motor learning.

While we see considerable value in this work in terms of expanding our conceptualization of the functional domain of the cerebellum, there are some methodological details that need to be clarified, as well as additional control analyses to support the interpretations of the imaging results, as well as further efforts in linking the current results to the literature.

Essential revisions:

1) Did the authors make a priori predictions about ROI activation? Given the work that has been done to map the cognitive cerebellum, the authors could have theorized that the activation would be elicited in "emotional" (Guell et al., 2018) or "prediction" (Moberget et al., 2014) regions of the cerebellum. Instead, their interpretations of the specific regional activations found seemed to be done a posteriori. This important distinction should be discussed. Moreover, several of the cerebellar foci reported in the current paper (Crus I, lobule VI) appear to overlap with cerebellar regions typically associated with higher cognitive function (e.g., working memory). Since the subjects quickly grasped the association between CS+ and US, and could in most cases explicitly report this afterwards, could representations of the learned associations in working memory explain the engagement of more "cognitive" cerebellar regions? In other words, if the association had been harder to detect explicitly by the subjects (e.g., by including many more conditioned stimuli with varying association levels with the US), would the authors expect the same results/regions?

2) Re: heart rate and respiration. Particularly with a fear conditioning paradigm, it could be argued that the results are vulnerable to task-related changes in heart rate, which can confound functional data. Importantly, both of these variables were measured and accounted for in the GLM. However, it would be useful to know if either one changed predictably during the acquisition and extinction phases (e.g., ramped up in anticipation of the US), so the potential for collinearity would be clearer.

3) One concern is that the PPI results (specifically the observed FC between cerebellum and occipital regions) could be driven by "bleed over" from visual cortex. The authors mention that they manually corrected the cerebellar mask but they did not provide any specific details. Given the importance of this analysis, extra care should be taken here to ensure that there is no mixture of signals. The mask could be recomputed (e.g., perhaps by regressing out signal from visual cortex like in Buckner et al. (2011) or a "buffer mask" could be created by removing voxels from both the occipital lobe + anterior cerebellum so that there is no abutting regions). One or both of these analyses used to recompute would provide assurance that the PPI results are not contaminated by bleed over.

4) One reviewer expressed concern that smoothing was done, given the high-resolution 7T data. While smoothing is common, it would be nice to know how different the (cerebellar) results come out if the data are unsmoothed.

5) Co-registration appeared to be done between the mean EPI and structural images. The EPIs were then corrected for slice timing and realigned to the first volume of the habituation phase. Were the EPIs first realigned to the "corrected" mean EPI image (the one coregistered with the anatomical image)? If not, it seems that the functional and anatomical images were not in the same space, which would be problematic. Moreover, the functional volumes were normalized to SUIT space, though it's not clear whether the mask used to do this was based only on the voxels from the corrected anatomical mask (cerebellum only) or whether it was a "functional" mask. This should be made clear in the manuscript.

6) The 8 second interval between the CS and US is much longer than what is thought to be the "effective" time scale of cerebellar learning (e.g., around 500ms in eyeblink conditioning paradigms; Schneiderman and Gormezano, 1964). The authors should discuss the choice of this particular interval, and address the discrepancy between the length of their chosen interval and the conventional understanding that there is only a small temporal window for cerebellar-driven learning to occur.

7) For the CS+>CS- contrast, was this analysis collapsed over the habituation, acquisition, and extinction phases? It would seem odd to include the habituation phase. Moreover, the acquisition and extinction phases should theoretically produce different results as well. The authors should perform this contrast over each phase separately. Moreover, it would be useful to have the CS+/CS- β results for the habituation phase (subsection “Mean β values related to each event (presentation of US, CS+, CS-, omission of US) compared to rest”) included, and those should also be included in Figure 5.

---

## [Author Response]

Essential revisions:1) Did the authors make a priori predictions about ROI activation? Given the work that has been done to map the cognitive cerebellum, the authors could have theorized that the activation would be elicited in "emotional" (Guell et al., 2018) or "prediction" (Moberget et al., 2014) regions of the cerebellum. Instead, their interpretations of the specific regional activations found seemed to be done a posteriori. This important distinction should be discussed.

The reviewers are correct in assuming that a nonselective analysis of cerebellar activations has been performed. Interpretation of specific regional activations has been done a posteriori.

We agree with the reviewers that another approach would have comprised a selective analysis of (a priori) functionally defined ROIs based on the recent studies mapping the cerebellum to a broad range of functions. Although this is a very attractive approach, there are still some limitations. Firstly, it remains unclear whether or not there is “a” prediction region in the cerebellum. This does not exclude that there is overlap between prediction-related activities across different tasks. A future meta-analysis of prediction-related cerebellar fMRI activity across different task domains would help to answer this question. Secondly, there are important open questions regarding cerebellar areas involved in emotions. The emotional area described by Guell et al. (2018) is based on one single task, but not on a meta-analysis of a significant number of different emotional tasks. In the King et al. (2019) study, four tasks included emotional processing. However, emotional processing did not become part of any of the ten main functional regions of the cerebellum introduced by the authors. Furthermore, they found that vermal activation was best explained by eye movements (or visual attention), but not emotional processing as reported by Guell et al. (2018).

Thus, as yet, we feel that a nonselective analysis gives a more unbiased picture of cerebellar regions involved in fear conditioning. We do acknowledge, however, that this approach does allow only limited conclusions on the possible contributions of selective cerebellar areas to learned fear responses. The Discussion has been revised accordingly (subsection “Cerebellar activation during the presentation and the prediction of aversive events”).

Moreover, several of the cerebellar foci reported in the current paper (Crus I, lobule VI) appear to overlap with cerebellar regions typically associated with higher cognitive function (e.g., working memory). Since the subjects quickly grasped the association between CS+ and US, and could in most cases explicitly report this afterwards, could representations of the learned associations in working memory explain the engagement of more "cognitive" cerebellar regions? In other words, if the association had been harder to detect explicitly by the subjects (e.g., by including many more conditioned stimuli with varying association levels with the US), would the authors expect the same results/regions?

This is a very good point. Yes, we fully agree that cerebellar activation in lobules VI and Crus I overlap with areas in the cerebellum which have been shown to contribute to cognitive functions including working memory (e.g. Guell et al., 2018; King et al., 2019, although in this study “active maintenance” was lateralized to the right; and visual working memory was localized in the vermis). We believe that this makes very good sense: in human studies, the main output measure of conditioned fear are increases in skin conductance response (SCR) and changes in ratings of US expectancy or fear (e.g. Lonsdorf et al., 2017 for recent review). A prerequisite for these conditioned fear responses to occur is that participants become aware of the CS-US contingencies (Dawson and Furedy, 1976; Lonsdorf et al., 2017, for reviews; Tabbert et al., 2006; Tabbert et al., 2011). Contingency awareness is likely linked to working memory processes (Dawson and Furedy, 1976). Thus, at least part of the activations seen in the lateral cerebellar hemisphere may be due to higher cognitive processes involved in differential fear conditioning. If this assumption is true, the pattern of cerebellar activation should be different in aware and unaware participants. This hypothesis could be tested in future studies, e.g. by using multiple CSs as suggested by the reviewers (see also Rehbein et al., 2014) or by using distractor tasks (e.g. Tabbert et al., 2006; Tabbert et al., 2011) to prevent awareness of the CS-US contingencies. The contribution of higher cognitive function to fear conditioning is discussed in more detail in the revised Discussion subsection “Cerebellar activation during the presentation and the prediction of aversive events”.

2) Re: heart rate and respiration. Particularly with a fear conditioning paradigm, it could be argued that the results are vulnerable to task-related changes in heart rate, which can confound functional data. Importantly, both of these variables were measured and accounted for in the GLM. However, it would be useful to know if either one changed predictably during the acquisition and extinction phases (e.g., ramped up in anticipation of the US), so the potential for collinearity would be clearer.

This is correct. In fear conditioning, changes of heart (and respiration) rate are to be expected. Both conditioned bradycardia and tachycardia has been observed, with conditioned bradycardia being more likely in a restraint animal, and in humans using neutral CSs (Lonsdorf et al., 2017; Maschke et al., 2002). To evaluate fear-conditioned variations in heart rate, however, an electrocardiogram (ECG) with a sampling rate of 500 Hz or above is recommended (Quintana et al., 2016). Unfortunately, a 7T fMRI compatible ECG was not available to us. Instead, heart rate was acquired with a pulse oximeter sensor attached to a finger with a sampling rate of 50 Hz. The low sampling rate, as well as the broad and time delayed pulse oximeter curve (compared to the ECG's well-defined and instantaneous R-peaks) did not allow for the accuracy needed to observe small variations in heart rate. This limitation has been added to the revised Materials and methods section (subsection “Physiological data acquisition”). Of note, an unpleasant, but not painful US was used, and only small changes in heart rate are to be expected. In fact, we did analyze the pulse oximetry data, and – as expected – were unable to detect any changes.

Changes in respiration rate are not recommended as a physiological outcome measure of fear conditioning in human studies (Lonsdorf et al., 2017). Given a normal respiration rate of 12-18/minute and a CS-US time window of 8 seconds, fear conditioned changes of respiration rate cannot be reliably assessed. In addition, the same limitations as for measures based on pulse oximetry apply.

For fMRI with a TR of 2 seconds, however, the measured pulse (and respiration) signal is of acceptable quality and can be used to perform retrospective physiological denoising (RETROICOR, Glover et al., 2000). For illustration, non-denoised evaluation of the three main contrasts in acquisition is now presented in Figure 3—figure supplement 2A-C.

3) One concern is that the PPI results (specifically the observed FC between cerebellum and occipital regions) could be driven by "bleed over" from visual cortex. The authors mention that they manually corrected the cerebellar mask but they did not provide any specific details. Given the importance of this analysis, extra care should be taken here to ensure that there is no mixture of signals. The mask could be recomputed (e.g., perhaps by regressing out signal from visual cortex like in Buckner et al. (2011) or a "buffer mask" could be created by removing voxels from both the occipital lobe + anterior cerebellum so that there is no abutting regions). One or both of these analyses used to recompute would provide assurance that the PPI results are not contaminated by bleed over.

Done. Data has been recomputed using the method introduced by Buckner et al. (2011). Data is presented in Figure 6—figure supplement 1. The general pattern did not change, so we do not have any hint for a possible “bleed over”.

4) One reviewer expressed concern that smoothing was done, given the high-resolution 7T data. While smoothing is common, it would be nice to know how different the (cerebellar) results come out if the data are unsmoothed.

Done. Additional data analysis of the three main contrasts is provided in Figure 3—figure supplement 2G-I, based on un-smoothed functional data. By omitting smoothing, cerebellar activation patterns are essentially unchanged, except for a small decrease of activation in the vermal region and lobule VIII.

5) Co-registration appeared to be done between the mean EPI and structural images. The EPIs were then corrected for slice timing and realigned to the first volume of the habituation phase. Were the EPIs first realigned to the "corrected" mean EPI image (the one coregistered with the anatomical image)? If not, it seems that the functional and anatomical images were not in the same space, which would be problematic. Moreover, the functional volumes were normalized to SUIT space, though it's not clear whether the mask used to do this was based only on the voxels from the corrected anatomical mask (cerebellum only) or whether it was a "functional" mask. This should be made clear in the manuscript.

We thank the reviewers for this helpful comment. The Materials and methods section has been revised to more correctly represent the actual order of preprocessing operations and the origin of the cerebellar mask used for normalization to SUIT space (subsection “Image processing”, last paragraph).

In brief, functional data was first slice-time corrected and realigned to the first volume of the habituation phase, by which step a mean functional image was auto-generated by SPM. This mean functional image was coregistered with the anatomical image and the coregistration transformation was subsequently applied to all the participants’ slice-time corrected and realigned functional volumes.

Cerebellar masks were generated using the SUIT Segment and Isolate routine, manually corrected by an experienced technician, slightly smoothed (isotropic kernel of 1.5 mm on 1 mm masks) to remove stray voxels and artificially steep steps in the mask outline from manual slice-wise drawing, thresholded against a value of 0.5 and supplied to the SUIT Normalize DARTEL routine.

6) The 8 second interval between the CS and US is much longer than what is thought to be the "effective" time scale of cerebellar learning (e.g., around 500ms in eyeblink conditioning paradigms; Schneiderman and Gormezano, 1964). The authors should discuss the choice of this particular interval, and address the discrepancy between the length of their chosen interval and the conventional understanding that there is only a small temporal window for cerebellar-driven learning to occur.

This is a very good question. Because measures of autonomic fear responses evolve slowly (SCR, heart rate change), fear conditioning in both animals and humans are commonly done with long CS-US intervals. Different to eyeblink conditioning, fear conditioning is tolerant to long CS-US periods (Carter et al., 2003; Lonsdorf et al., 2017). This allows to perform event-related fMRI studies, which can separate between CS-related activation, US-related activation, and activation related to the omission of the US – which is not possible in eyeblink conditioning studies (using the most optimal CS-US time window of around 500 ms). Both animal (Lavond et al., 1984; Sacchetti et al., 2002; Supple and Leaton, 1990a; Supple and Leaton, 1990b; Supple and Kapp, 1993) and human studies (Maschke et al., 2002) have shown that the cerebellar vermis is involved in fear conditioning – despite these long CS-US time intervals. Read-outs in theanimal studies were either freezing behavior or changes in heart rate. We agree with the reviewers that this is somewhatcounterintuitive, because the cerebellum is generally thought to be important in tasks with much smaller time scales. The contribution of the cerebellum to discrete somatic motor behavior (such as eyeblink responses) may be different compared to slowly reacting autonomic responses, but also in cognitive tasks (e.g. working memory or language tasks). We added a brief discussion of the differences between eyeblink and fear conditioning to the revised manuscript to account for this helpful comment (subsection “Fear conditioning”, last paragraph).

7) For the CS+>CS- contrast, was this analysis collapsed over the habituation, acquisition, and extinction phases? It would seem odd to include the habituation phase. Moreover, the acquisition and extinction phases should theoretically produce different results as well. The authors should perform this contrast over each phase separately. Moreover, it would be useful to have the CS+/CS- β results for the habituation phase (subsection “Mean β values related to each event (presentation of US, CS+, CS-, omission of US) compared to rest”) included, and those should also be included in Figure 5.

The fMRI contrasts displayed were collapsed over early and late blocks, but not across experimental phases. Materials and methods have been revised accordingly for clarification (subsection “fMRI analysis”, third paragraph). Results for habituation phase have now been included in the results presented in Table 1; there were no significant activations observed. CS+/CS- β values for the habituation phase have been included in the revised Figure 5.